# You Only Write Thrice:
# Creating Documents, Computational Notebooks and Presentations From a Single Source

**Kacper Sokol** and **Peter Flach**
Department of Computer Science,
University of Bristol,
Bristol, United Kingdom
{K.Sokol,Peter.Flach}@bristol.ac.uk

## Abstract

Academic trade requires juggling multiple variants of the same content published in different formats: manuscripts, presentations, posters and computational notebooks. The need to track versions to accommodate for the write–review–rebut–revise life-cycle adds another layer of complexity. We propose to significantly reduce this burden by maintaining a single source document in a version-controlled environment (such as git), adding functionality to generate a collection of output formats popular in academia. To this end, we utilise various open-source tools from the Jupyter scientific computing ecosystem and operationalise selected software engineering concepts. We offer a proof-of-concept workflow that composes Jupyter Book (an online document), Jupyter Notebook (a computational narrative) and reveal.js slides from a single markdown source file. Hosted on GitHub, our approach supports change tracking and versioning, as well as a transparent review process based on the underlying code issue management infrastructure. An exhibit of our workflow can be previewed at

https://so-cool.github.io/you-only-write-thrice/.

## 1 Source Multiplicity

Despite immense technological advances permeating into our everyday life and work, scientific publishing has seen much less evolution. While moving on from hand- or type-written manuscripts to electronic documents allowed for faster and more convenient editing, dissemination and review, the (now obsolete and unnatural) limitations of the "movable type" publication format persist. Among others, this glass wall poses significant challenges to effective communication of scientific findings with their ever-increasing volume, velocity and complexity. To overcome these difficulties we may need to depart from the, de facto standard, *Portable Document Format* (PDF) and set our sight on something more flexible and universal. In the past three decades the World Wide Web has organically evolved from a collection of hyper-linked text documents into a treasure trove of diverse and interactive resources – a process that should inspire an evolution of the scientific publishing workflow.

A physical, printed and bound, copy of a research paper may have been mostly replaced by a PDF document viewable on a wide array of electronic devices, but both formats effectively share the same limitations. These constraints have recently become even more prominent with research venues requiring to publish supplementary materials that fall outside of the textual spectrum. Releasing *source code* is advised to ensure reproducibility; *computational notebooks* highlight individual methods, experiments and results; recording a short *promotional video* helps to advertise one's research; *slides* and *oral presentations* (often pre-recorded nowadays) disseminate findings and foster discussions; and *posters* graphically summarise research contributions. Notably, these artefacts are usually created independently and are distributed separately from the underlying paper – nonetheless while disparate in appearance, form and function, they all share a common origin.

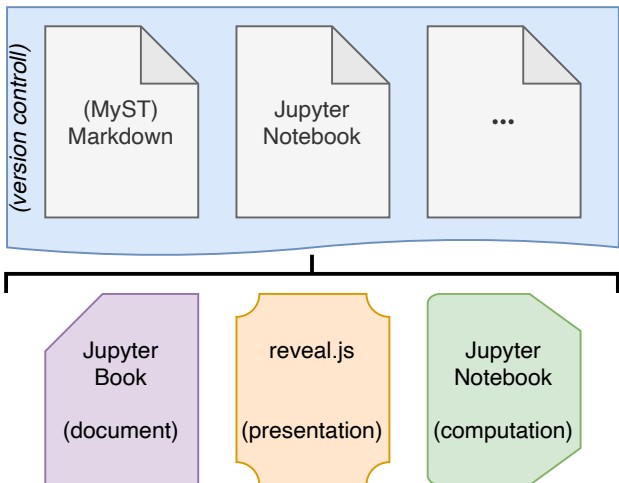

Figure 1: Generating multiple conference artefacts such as documents (papers), presentations (slides) and computational notebooks from a single source repository can streamline the academic publishing workflow.

Without a designated place for each research output and a dedicated workflow to create and distribute them, they may end up scattered through the Internet. Code may be published on GitHub, computational notebooks distributed via Google Colab or MyBinder, and videos posted on YouTube. Slides and posters, on the other hand, tend to accompany the paper – which itself is placed in proceedings – sparingly with an often outdated version of the article available through arXiv. While organising, distributing and linking all of these resources is a goal in itself, we shall first reconsider the authoring process responsible for their creation. Since delivering each format appears to be an independent task that requires a dedicated effort, having as many of these resources as possible generated from a single source could streamline the process and create a coherent narrative hosted in a single place. While as a community we created technology powerful enough to train vision models in a web browser[1], we find it somewhat difficult to see beyond the restrictive PDF format and hence increasingly lack publishing tools necessary to carry out our work effectively.

To streamline the process of creating academic content we propose to generate conference artefacts – such as documents (papers), presentations (slides) and computational notebooks – from a single source as depicted in Figure 1. By storing all the information in a version-controlled environment (e.g., `git`) we can also track article evolution and revision, and facilitate a review process similar to the one deployed in software engineering and closely resembling the OpenReview workflow. Combining together research outputs as well as their reviews and revisions could improve credibility and provenance of scientific finding, thus simplifying re-submission procedures and taking the pressure of overworked reviewers. Additionally, including *interactive* materials – such as code boxes and computational notebooks – in the published resources encourages releasing *working* and accessible code.

The envisaged workflow generalises beyond academic publishing and may well be adopted for teaching, for example, producing lecture notes, slides and (computational) exercise sheets. Our proof of concept consists of:

- documents based on *Jupyter Book* (Executable Books Community, 2020);
- computational narratives presented as *Jupyter Notebooks* (Kluyver et al., 2016); and
- presentations (decks of slides) created with *reveal.js* (El Hattab, 2020).

A self-contained example of generating these three artefacts from a single source document is published with GitHub Pages and accessible at

`https://so-cool.github.io/you-only-write-thrice/.`

---

[1] `https://teachablemachine.withgoogle.com/`

The text is written in extended markdown syntax (MyST flavour), and the code is executed in Python, however any programming language supported by the Jupyter ecosystem can be used. These sources are hosted on GitHub and can be inspected at



`https://github.com/so-cool/you-only-write-thrice/.`



Publishing these diverse materials as interactive web resources democratises access to cutting-edge research since they can be explored directly in the browser without installing any software dependencies. While the main output of the proposed workflow is a collection of HTML pages, it can also produce static formats such as PDF or EPUB, albeit forfeiting all of the discussed advantages.

As it stands, our workflow consists of open-source tools, but it relies on free tiers of commercial services. To ensure longevity and sustainability, we need community-driven software and infrastructure for hosting, reviewing and publishing resources in the proposed formats, for example, taking inspiration from the OpenReview model. Being able to influence its development, we could prioritise features needed for a wider adoption of this bespoke platform, e.g., by implementing anonymous submission and review protocols, which are not available in commercial solutions such as GitHub or Bitbucket. At the content-generation end, our workflow relies heavily on the Jupyter ecosystem. Jupyter Book and MyST Markdown provide basic text formatting and implement academic publishing features such as mathematical typesetting (with LaTeX syntax) and bibliography management (with BIBTeX). The platform is still in early development and exhibits certain limitations, e.g., fine-grained layout customisation may be difficult, which is particularly noticeable with reveal.js slides. However, the Jupyter Book environment can be extended with custom plugins, which in the long term may be as plentiful as LaTeX packages; for example, we built support for non-mainstream programming languages such as SWI Prolog[2], cplint[3] and ProbLog[4] that are not natively supported by Jupyter. While the openness and transparency of the proposed workflow can be considered its forte, the same qualities may also pose challenges for academic publishing, which need to be explored further before pursuing this avenue.

A part of our workflow derives from the *computational narrative* concept, which interweaves prose with executable code, thus improving reproducibility and accessibility of scientific findings. The most prominent example of this technology are Jupyter Notebooks delivered to the audience through MyBinder or Google Colab. Content that is more narrative-driven, on the other hand, can be published with Bookdown (R Studio, 2020) – a toolkit comparable to Jupyter Book but stemming from the R language ecosystem. A similar publication platform, dedicated to research papers, is Distill[5], however its wider adoption is hindered by the degree of familiarity with web technologies required of authors. Additionally, the proposed workflow borrows from software engineering; in particular, code versioning and review. The former is not widely adopted by the scientific community, partially contributing to scientific papers lacking evolution traces and provenance that could shine a light on their journey through rejection and acceptance at various workshops, conferences and journals. On the other hand, the software-like review process of academic writing has been adopted by The Open Journals[6], for example, the Journal of Open Source Software[7], further improving on the model operationalised by OpenReview.

## 2  PUBLISHING WORKFLOW

The academic publishing life-cycle consists of four distinct steps: composing, reviewing, publishing and presenting. Researchers need to write down their findings, get them reviewed and revised, formally publish them adhering to bibliometric standards, and, finally, present their work in different formats: as academic writing, conference talks, poster sessions, promotional videos, blog posts,

---

[2]`https://book-template.simply-logical.space/`

[3]`http://cplint-template.simply-logical.space/`

[4]`https://problog-template.simply-logical.space/`. ProbLog is unique in this aspect as it can either be executed directly from Python (through a custom interpreter, thus not requiring a dedicated plugin) or with bespoke code boxes (as is the case with SWI Prolog and cplint).

[5]`https://distill.pub/`

[6]`https://www.theoj.org/`

[7]`https://joss.theoj.org/`. The submission and review process is outlined in the journal's documentation published at `https://joss.readthedocs.io/`.

press releases and public engagement events, among others. Each artefact produced in this process constitutes a different entry point to the research outputs, allowing a diverse audience to freely explore a wide range of its – theoretical, computational and experimental – aspects. However, despite sharing a common origin, the current publishing workflow requires us to craft each piece of this mosaic separately.

To address this issue, we propose a proof-of-concept pipeline for composing academic articles, computational notebooks and presentations from a single curated source, helping to document and disseminate research in accessible, transparent and reproducible formats. To this end, we harness modern web technologies, e.g., reveal.js, and open-source software from the Jupyter ecosystem, in particular Jupyter Book and Jupyter Notebook/Lab. Such an approach allows exploring and interacting with research outputs directly in a web browser (including mobile devices), thus alleviating technological and software barriers – akin to what PDF did for electronic documents. Finally, we tackle the review step by linking it to the document source, which permits a much more structured and conversational process that feels more natural and intuitive.

**Composing**    The backbone of our workflow are documents written in MyST Markdown – an extended markdown syntax that supports basic academic publishing features such as tables, figures, mathematical typesetting and bibliography management. Content written in this format is highly interoperable and can be converted to LaTeX, HTML, PDF or EPUB outside of the proposed system, thus serve as a source for or a component of other authoring environments. The key to automatically composing a variety of output formats is the syntactic sugar allowing to superficially split the content into *fragments* and annotate them. These *tags* prescribe how each piece of prose, figure or code should be treated – e.g., included, skipped or hidden – when building different target formats. Then, Juptyer Book can process selected fragments to generate web documents and computational narratives, the latter of which may be launched as Jupyter Notebooks with either MyBinder or Google Colab. While this is already a step towards "reproducibility by design", having content that depends on code implicitly encourages releasing it as a software package that can be invoked whenever necessary, therefore improving reproducibility even further. The aforementioned source annotation can also specify a slide type and its content in a presentation composed with reveal.js, streamlining yet another aspect of the academic publishing workflow. While each of these artefacts is destined for web publication, their trimmed version can also be exported to formats such as PDF or EPUB.

**Reviewing**    In the proposed workflow, we envisage storing the document source in a version-controlled environment similar to `git`, which has two benefits. First, it enables tracking changes in the document, versioning it and monitoring its evolution through various workshops, conferences and journals submissions. Secondly, such a setting supports peer review inspired by code review in software engineering. In this model, the reviewers could attach their comments to specific locations in the paper, allowing other reviewers to chime in and the paper authors to address very specific concerns explicitly linked to the submitted document. Furthermore, this structure creates a discussion-like experience, which should feel more natural to humans – akin to comments and discussions in shared document writing platforms such as Google Docs, Microsoft Office Word and Overleaf. The entire process can be made more structured and objective by providing the reviewers with general checklists and a list of tags to annotate each of their concerns (e.g., typo, derivation error or incorrect citation). The rebuttal and revision stage is also simplified in this framework since all of the changes applied to the document are tracked and can be linked to individual reviewer comments.

Such a formalisation of the review–rebut–revise cycle significantly increases the transparency and provenance of the entire process. This approach can be trialled through the *Pull Request* functionality of commercial code sharing platforms, such as GitHub or Bitbucket, before investing more time and resources into the development of a dedicated (self-hosted and open-source) technology. While doing so would not allow for anonymous peer-review, the process could start with implementing and improving upon the aforementioned model operationalised by the the Open Journals, helping to identify and prioritise features expected of the dedicated platform. Similarly, displaying a reviewer's comments could be delayed until the review is finished to avoid bias, followed by merging them with other comments placed in close proximity. Notably, adapting the proposed review format would not require version-control or software engineering skills since all of the complexities are abstracted away by the user interface. Since the review could be permanently attached to the submission, the

implications of this approach would need to be studied and understood before enforcing it. Alternatively, or in addition to the above process, external services such as *hypothesis*[8] or *utterances*[9] – which are available as (experimental) Juptyer Book plugins – could be used to collaboratively review, comment, discuss or annotate submissions.

**Publishing**  Since the content source is stored in a version-controlled environment, one can imagine submitting a document for review by specifying its particular version (e.g., by tagging a selected `git` commit), with the publication process following the same procedure. Such a versioning approach would also demystify the journey of a paper through various workshops, conferences and journals, and clarify the improvements made after each rejection. In this setting, bibliometrics can be achieved by automatically minting Digital Object Identifiers (DOI) upon publication, for example, using *zenodo*[10], which is already integrated with software versioning mechanisms provided by GitHub and Bitbucket. Another bibliometrics strategy suitable for web technologies can be derived from tools such as Google Analytics, which could be deployed to collect fine-grained information about the readers and hyper-links pointing to and from the publication, thus allowing to build a detailed network of connected documents. While the format is intended for web publication, it can also be stored on a personal computer or converted into monolithic entities such as LaTeX, PDF or EPUB. This interoperability allows to archive any or all variants of the document to ensure its longevity and accessibility. Notably, by connecting the local copy to a custom execution environment, the interactivity of the materials can be preserved offline.

**Presenting**  The proposed authoring framework alleviates the need to create separate articles, computational narratives and slides by building them from a single markdown source. Since these artefacts are intended for web publication, they can take advantage of modern technologies that can make them interactive, thus more engaging. For example, the RISE extension of the Jupyter Notebook platform (Avila, 2020) allows launching reveal.js presentations with executable code boxes. By building bespoke plugins, we can enable support of less prominent programming languages (recall the aforementioned example of SWI Prolog) and create additional output formats such as blog posts or academic posters. Since the materials are delivered as web pages, technological barriers are lifted, portability is guaranteed and sharing is made easy. Finally, the proposed workflow can be deployed in education to prepare lectures, courseworks, exercises, notes and (self-)study materials, therefore supporting both synchronous and asynchronous learning – see the pre-release[11] of our interactive edition (Flach & Sokol, 2018) of the *Simply Logical* textbook (Flach, 1994) for an example.

## 3 CONCLUSIONS

In this paper we proposed a novel publication workflow built around Jupyter Book, Jupyter Notebook and reveal.js. Our exhibit demonstrates how to create narrative-driven documents (peppered with executable code examples), computational notebooks and interactive slides, all from a single markdown source. Furthermore, we outlined a strategy for hosting and disseminating such materials through version-controlled environments similar to code sharing repositories. Such a platform facilitates an intuitive review mechanism inspired by software engineering practice, thus endowing provenance and transparency to the scientific publication process. While our exhibit is currently a bare-bones proof-of-concept built from open-source tools, it shows the potential for transforming the current PDF workflow into an environment focused on content creators and reviewers. One can even imagine automating parts of this process with "bots" validating submissions based on predefined criteria, and partially pre-populating a review form to streamline the entire publication life-cycle (akin to continuous integration and deployment pipelines in software engineering).

While addressing some of the main issues with current publishing practice, the proposed workflow is not (yet) a silver bullet and the underlying technology needs further development to mature into reliable software. We envisage that engagement of the scientific community and open discussion are needed to steer the development and foster broader adoption of such tools, for example, workshops

---

[8]https://hypothes.is/
[9]https://utteranc.es/
[10]https://zenodo.org/
[11]https://too.simply-logical.space/

encouraging submission and review in such a format. Interactivity is a great advantage of Jupyter Book publications, but the compute resources employed to execute the underlying code have to be accounted for and provisioned since relying upon free code-execution environments (such as Google Colab and MyBinder) is not sustainable in the long term. Given that the main output of the proposed workflow is a collection of web pages, they either need to be accessed online or downloaded and browsed locally to take full advantage of their format; generating PDFs and EPUBs is also possible, however they lack interactivity and may not be visually appealing since they only play a secondary role. Notably, the proposed framework is not as powerful as LaTeX, which benefits from decades of development and a rich ecosystem of packages, therefore any customisation or automation will require a bespoke plugin that may be slow to create and buggy at first. Nonetheless, without making the first step – and addressing the disadvantages associated with it – the publishing process will not benefit from the technology that we developed to make our research possible in the first place.

### ACCESSIBILITY STATEMENT

The main output format of the proposed Jupyter Book publishing workflow are HTML pages, and the ubiquity of web technologies ensures availability of compatible accessibility tools. In particular, adherence to web accessibility standards and utilisation of commonplace assistive technology can render the artefacts of our workflow widely approachable. For example, tools such as navigation aids; screen readers; screen magnifiers; page display colour, contrast and brightness adjustments; as well as font (size and colour) customisation can be easily deployed. Additionally, accessibility of computational notebooks is constantly monitored and improved by the Project Jupyter's Accessibility Working Group[12]. Similarly, the reveal.js platform allows inclusion of custom plugins, with availability of community-developed accessibility improvements[13].

Since the generated web resources are *static* (from the technology perspective), they can be downloaded onto a personal computer and browsed offline. The use of modern web technologies makes them *responsive*, thus compatible with computers and mobile devices of varying screen size. In addition to the HTML output, the proposed workflow can build other popular formats such as PDF or EPUB, which are suitable for tablets and e-readers. We envisage hosting the content source files at popular code sharing and versioning platforms – such as GitHub or Bitbucket – that support rich ticketing and issue management systems. In particular, the readers can use these mechanisms to provide feedback and raise accessibility concerns.

### ACKNOWLEDGEMENTS

This work was supported by the TAILOR Network[14] – an ICT-48 European AI Research Excellence Centre funded by EU Horizon 2020 research and innovation programme, grant agreement number 952215. Among others, TAILOR explores novel ways of working and publishing, including AI-powered collaboration tools, and AI training platforms and materials.

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
