# OpenReview forum: "You Only Write Thrice: Creating Documents, Computational Notebooks and Presentations From a Single Source"
_ICLR.cc/2021/Workshop/Rethinking_ML_Papers/Exhibit_and_Workflow — Rethinking ML Papers - ICLR 2021 workshop Oral_

### Official Review · Reviewer_eQNu · 2021-03-30
**Review: You only write thrice**

**Accessibility:**

Score of 5 (Exceptional): Submission identifies and articulates accessibility matters, provides justifications for the proposed paradigm, and declares the limitations.

**Litreview:**

Score of 4 (Strong): The submission directly differentiates itself from previous works and formats.

**Problemstatement:**

Score of 4 (Strong): The submission sets a very strong example of how to address the problem, which should be relevant to the workshop themes.

**Relevance:**

Score of 5 (Exceptional): Like (4) but does so with multiple themes of the workshop.

**Results:**

Score of 5 (Exceptional): Submission has an excellent design and all criteria are addressed. Conclusions, practical/theoretical implications are well articulated.

**Reviewerconfidence:**

5. The proposed framework is novel in that the closest frameworks could be Github pages or the OpenReview site (with links to the different materials). However, what the authors propose is to have a workflow through a single source file that helps with hosting, versioning, and potential reviewing/tracking.

**Reviewtext:**

This paper involves creating a proof-of-concept framework for hosting/publishing manuscripts as well as any supplementary material such as presentations/posters/source code which ensures reproducibility and allows authors to explain results. The authors realize that while the different material "share a common origin" they are usually hosted on different sites (Google Colab for code, Youtube for videos, and usually arxiv for preprints) making it a hassle to review the content. They propose an unified framework built on open-source tools consisting of Jupyter Book text, Jupyter notebook code, and reveal.js slide decks (with a demo) which enhances accessibility and ease of use with the additional benefit of making reviewing easier (in the form of tags/annotations/comments) and the potential of version-control.

**Score:**

Strong accept: The reviewer has a strong enthusiasm to apply the proposed framework in their work.

---

### Official Review · Reviewer_YogX · 2021-04-01
**You only Write Thrice Review**

**Accessibility:**

Score of 5 (Exceptional): Submission identifies and articulates accessibility matters, provides justifications for the proposed paradigm, and declares the limitations.

**Litreview:**

Score of 4 (Strong): The submission directly differentiates itself from previous works and formats.

**Problemstatement:**

Score of 5 (Exceptional): The submission states a well-known problem relevant to the workshop, and sets what could be a new standard in the field when it comes to addressing it.

**Relevance:**

Score of 4 (Strong): The submission directly addresses a theme of the workshop, and does so in a very professional manner.

**Results:**

Score of 4 (Strong): Submission is very well structured and follows all the criteria (i.e. clarity, novelty, interactivity, and coherency). However, practical significance/theoretical implications are not discussed.

**Reviewerconfidence:**

4/5: The authors have a compelling idea. Further thought around the implications of this tool should lead to interesting conversation at the workshop!

**Reviewtext:**

The authors propose a tool for authoring scientific research in multiple forms. The problem and solution is compelling as researchers do create multiple documents and artefacts. I appreciate how the authors think about better supporting researchers as publication authors. A topic I hope to see more thoroughly addressed is how the authors envision tools and workflows such as the one they propose would change researchers’ relationships to the public. Specifically, I think about how researchers may consider themselves to be “public scholars.” As such, they spend a significant amount of creativity and time thinking about how to adapt their research findings and communication to different groups and different formats. Part of this process in return informs their research or intellectual curiosity. It seems that duplicity is inherent to and arguably essential to gain these benefits. What do the authors think are the benefits and drawbacks of their approach?

The paper should facilitate interesting conversation with potential users!


**Score:**

Accept: The reviewer believes the submission provides a novel and reliable scheme to improve science communication but needs improvement.

---

### Meta-Review · Program_Chairs · 2021-04-01

**Recommendation:** Accept
**Confidence:** 5

**Metareview:**

The authors present a compelling case study describing a novel system for literate programming. Comparison with other similar publishing platforms is scarce and would be a welcome addition.

---

### Decision · Program_Chairs · 2021-04-01

Accept (Oral)